# Archaeoscape: Bringing Aerial Laser Scanning Archaeology to the Deep Learning Era

**Yohann Perron**[*] [1, 2]
yohann.perron@efeo.net

**Vladyslav Sydorov**[*] [1]
vladyslav.sydorov@efeo.net

**Adam P. Wijker** [1, 3]
adam.wijker@efeo.net

**Damian Evans** [†] [1]

**Christophe Pottier** [1]
christophe.pottier@efeo.net

**Loic Landrieu** [2]
loic.landrieu@enpc.fr

[1] École française d'Extrême-Orient (EFEO)     [2] LIGM, École des Ponts, CNRS, UGE

[3] Université Paris 1 Panthéon-Sorbonne

## Abstract

Airborne Laser Scanning (ALS) technology has transformed modern archaeology by unveiling hidden landscapes beneath dense vegetation. However, the lack of expert-annotated, open-access resources has hindered the analysis of ALS data using advanced deep learning techniques. We address this limitation with Archaeoscape (available at https://archaeoscape.ai/data/2024), a novel large-scale archaeological ALS dataset spanning $888 \text{ km}^2$ in Cambodia with 31,141 annotated archaeological features from the Angkorian period. Archaeoscape is over four times larger than comparable datasets, and the first ALS archaeology resource with open-access data, annotations, and models.

We benchmark several recent segmentation models to demonstrate the benefits of modern vision techniques for this problem and highlight the unique challenges of discovering subtle human-made structures under dense jungle canopies. By making Archaeoscape available in open access, we hope to bridge the gap between traditional archaeology and modern computer vision methods.

## 1 Introduction

Airborne Laser Scanning (ALS) has been celebrated as a "geospatial revolution" in modern archaeology due to its ability to penetrate vegetation and unveil traces of human activities that may otherwise be concealed or invisible [1, 2]. Extensive acquisition campaigns conducted in Southeast Asia [3], Central America [4], and Europe [5, 6] have led to a reevaluation of the historical impact of humans on "natural" landscapes, especially in tropical regions [7]. However, finding archaeological features in vast volumes of ALS data presents a significant challenge. Manual analysis is time-consuming and requires advanced expert knowledge of the studied civilization as well as on-site validation [8].

The emergence of deep learning offers a promising tool to assist researchers in identifying archaeological patterns, simplifying the exploration of these extensive acquisitions. Yet, the development of specialized models is hampered by the lack of expert-annotated datasets. In response, we introduce Archaeoscape, the largest open-access ALS dataset for archaeological research published to date. Spanning $888 \text{ km}^2$, it comprises 31,411 annotated instances of anthropogenic features of archaeological interest. The dataset includes orthophotos and LiDAR-derived normalized Digital Terrain Models

---

[*]Equal contribution.     [†]Posthumous authorship.

38th Conference on Neural Information Processing Systems (NeurIPS 2024) Track on Datasets and Benchmarks.

ALS 3D point cloud          Terrain model          Annotations

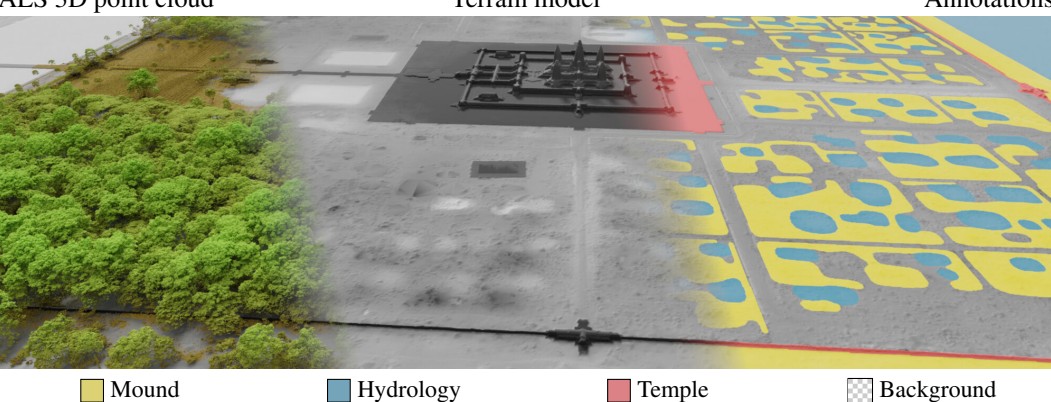

□ Mound          ■ Hydrology          ■ Temple          ▨ Background

**Figure 1: Archaeoscape.** Our proposed dataset contains 888 km$^2$ of aerial laser scans taken in Cambodia. The 3D point cloud LiDAR data (left) was processed to obtain a digital terrain model (middle). Archaeologists have drawn and field-verified 31,411 individual polygons by delineating anthropogenic features (right).

(nDTM), encompassing over 3.5 billion pixels with RGB values, nDTM elevation, and semantic annotations.

Traditionally, U-Net models [9] have dominated archaeological studies. In this paper, we evaluate several recent architectures for semantic segmentation. Our findings indicate that identifying ancient features beneath vegetation canopies using ALS still poses significant challenges. These difficulties can be attributed to the subtle nature of the objects sought, which are largely represented by faint elevation patterns. Moreover, certain features can span several kilometers and require extensive spatial context to disambiguate. With Archaeoscape, we aim to challenge the machine learning and computer vision communities to address the rich, impactful, and unsolved problem of ALS-based archaeology. At the same time, we also encourage the archaeological community to adopt open-access policies and explore modern deep learning approaches.

## 2 Related work

In this section, we explore the advantages of Archaeoscape over existing datasets, highlighting its larger scale and open-access policy (Section 2.1), and present the different models evaluated in our benchmark (Section 2.2).

### 2.1 ALS archaeology datasets

Deep learning for ALS archaeology is a dynamic field [10]. In Table 1, we list the main deep learning works on ALS-archaeology. Archaeoscape is not only one of the few open-access datasets available but also the largest and most comprehensively annotated by a significant margin. [1]

**Open-access policies.** ALS archaeology datasets typically withhold data, annotations, and code due to legitimate concerns about misuse [11, 12], and the absence of established open-access norms in archaeology. However, recognizing the critical role of reproducibility and open access in science, we make Archaeoscape accessible to academic researchers. We implement strict safeguards to protect sensitive archaeological information, as described in Section 3.1.

**Scope and extent.** Archaeoscape is the largest ALS archaeology dataset in terms of area covered (888 km$^2$) and number of annotated instances (31,411). Archaeoscape covers a 2× larger surface area and contains 3× more instances than the next-largest closed archaeology LiDAR dataset (see Table 1). It is also the first such dataset related to the Khmer civilization of Southeast Asia.

---

[1] We consider a dataset to be open-access when data, annotations, and train/test split are accessible, allowing the replication of the results.

**Table 1: ALS archaeology datasets.** Archaeoscape is the first open access ALS archaeology dataset to cover Southeast Asia, and to provide high resolution aerial photo imagery. It is also significantly more extensive than existing datasets in terms of surface area and number of annotated instances.

| | open-access | hi-res RGB | location | extent in km$^2$ | resolution in meters | number of instances |
|---|:---:|:---:|---|---|---|---|
| Arran [13] | ✔ | ✗ | United Kingdom | 25 | 0.5 | 772 |
| Litchfield [14] | ✗ | ✗ | USA | 50 | 1 | 1,866 |
| Puuc [15] | ✗ | ✗ | Mexico | 23 | 0.5 | 1,966 |
| AHN [16] | ✗ | ✗ | Netherlands | 81 | 0.5 | 3,553 |
| AHN-2 [17] | ✗ | ✗ | Netherlands | 437 | 0.5 | 3,849 |
| Connecticut [18] | ✗ | ✗ | USA | 353 | 1 | 3,881 |
| Dartmoor [19] | ✗ | ✗ | United Kingdom | 12 | 0.5 | 4,726 |
| Pennsylvania [20] | ✔ | ✗ | USA | 4 | 1 | 4,376 |
| Uaxactun [21] | ✗ | ✗ | Guatemala | 160 | 1 | 5,080 |
| Chactún [22] | ✗ | ✗ | Mexico | 230 | 0.5 | 10,894 |
| **Archaeoscape (ours)** | ✔ | ✔ | Cambodia | **888** | 0.5 | **31,411** |

## 2.2 Semantic segmentation with deep learning

ALS archaeology approaches rely predominantly on U-Net-based models [9]. However, the field of semantic segmentation has evolved considerably since its introduction in 2015. We propose to assess the performance of an array of contemporary, state-of-the-art models on the Archaeoscape benchmark. Models and pretraining strategies evaluated in Table 2 are denoted in **bold** throughout the text for clarity and ease of reference.

**Convolution-based models.** Convolutional Neural Networks (CNNs) [23, 24], and the **U-Net** [9] architecture in particular, remain the predominant choice for dense prediction tasks across various application fields due to their simplicity and effectiveness. **DeepLabv3** [25] improves on this model by using dilated convolution and Spatial Pyramid Pooling [26] to learn multiscale features.

**Vision transformers.** Vision transformers harness the versatility and expressivity of transformers [27] to extract rich image features. The Vision Transformer **ViT** [28] model splits the images into small patches, which are embedded with a linear layer, while the final patch encodings are converted into pixel prediction with another linear layer. **DOFA** [29] embed each input channel conditionally to its wavelength, allowing generalization to new sensors. Alternatively hybrid **HybViT** replaces these linear layers with a combination of convolutional and deconvolutional layers for encoding and decoding patches, respectively. This adaptation is particularly effective on smaller datasets, as the convolutions help capture local feature dependencies more effectively.

**Hierarchical ViTs.** Several variants of the ViT model use a hierarchical approach to effectively capture spatial features with a large context. The Pyramid Vision Transformer (PVT) [30] applies its attention mechanism according to a nested hierarchical structure, while **SWIN** [31] uses overlapping windows of increasing sizes. Building on these concepts, **PCPVT** [32] introduces a conditional relative position encoding mechanism, and **PVTv2** [33] also allows for overlapping patches.

**Pre-training strategies.** Recent advances in self- and weakly-supervised learning have profoundly impacted the efficacy of neural networks. These strategies often use large datasets with text annotations such as **CLIP-OPENAI** [34] or its open-source counterpart **CLIP-LAION2B** [35]. Alternatively, **DINOv2** [36] learns from large, unannotated image datasets. The recent Masked Auto-Encoder [37] tunes large models by using the pretext task of masked patch reconstruction. This approach has been adapted to address the specific needs of aerial imagery, leading to variants such as **ScaleMAE** [38] which are trained on satellite images.

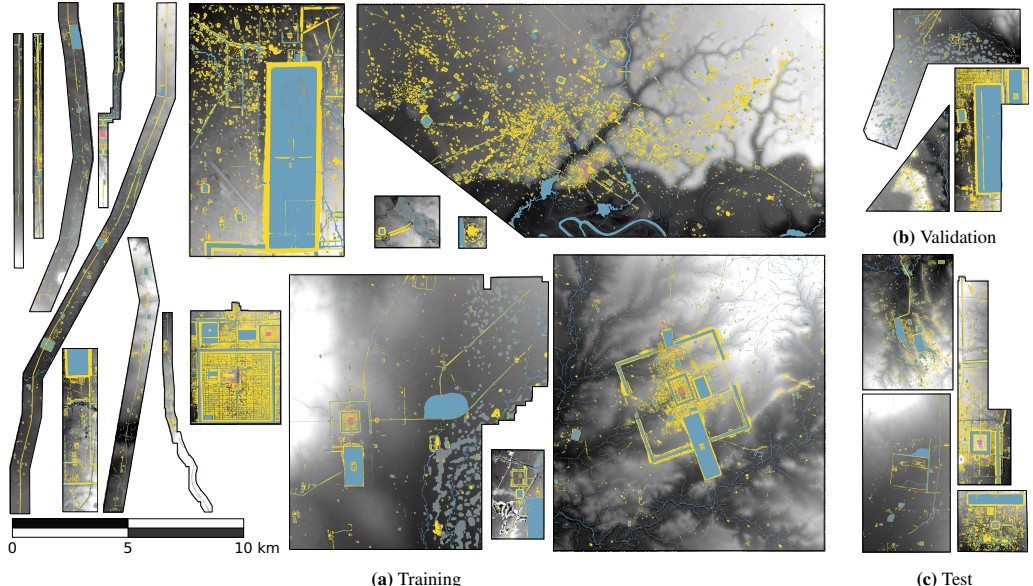

**(b)** Validation

**(a)** Training

**(c)** Test

**Figure 2: Archaeoscape overview.** We show the vectorial annotations overlaid onto the relative elevation maps for each parcel, and their assignment to the training, validation or test splits. The position and orientation of the parcels is arbitrary. The geometry of the annotations has been simplified to reduce the file size of the paper. Best viewed on a computer screen.

## 3 Archaeoscape

In this section, we describe the content of Archaeoscape (Section 3.1), as well as its acquisition process (Section 3.2).

**Context.** Angkor, the heart of the medieval Khmer Empire, is often referred to as a "hydraulic city" due to its extensive water management infrastructure. This system allowed the Khmer to thrive in a challenging environment, oscillating between monsoon and dry seasons, from the 9th to the 15th century. Today, much of the built environment of Angkor and the other cities of this period has disappeared, as virtually all non-religious architecture was built using perishable materials such as wood. What remains is often hidden by dense vegetation or damaged by erosion and modern agricultural practices, rendering these sites nearly invisible at ground level, so that even experts might walk over such sites without realizing it. However, the advent of LiDAR (Light Detection and Ranging) technology has been transformative, uncovering distinct, often geometric patterns in the topography indicative of ancient occupation and landscape alteration. By combining careful analysis of ALS imagery with targeted ground surveys, this decade-long project has documented tens of thousands of ancient Khmer features, many previously undiscovered, providing a new and expanded perspective on the history of the region.

### 3.1 Dataset characteristics

**Splits.** As shown in Figure 2, the dataset consists of 23 non-overlapping parcels of varying size, ranging from 2 to 183 km$^2$, and include archaeologically relevant areas such as ancient temples, cities, and roadways. We present the splits for Archaeoscape's training (623 km$^2$, 16 parcels), validation (97 km$^2$, 3 parcels), and test (168 km$^2$, 4 parcels) sets. The splits were chosen to respect the global distribution of features and landscapes: densely or scarcely occupied regions, hills or floodplains, large-scale hydraulic engineering sites, monumental temples, and *subtle* earthen features.

Under these constraints, splitting the dataset into spatially distinct regions—as is commonly done in geospatial machine learning—proved impractical. To prevent data contamination all parcels are separated by a least a 100 meter buffer. The test set consists of 2 *remote* parcels, set apart from the others by more than 5 km, and 2 parcels *adjacent* to training and validation sets, covering two

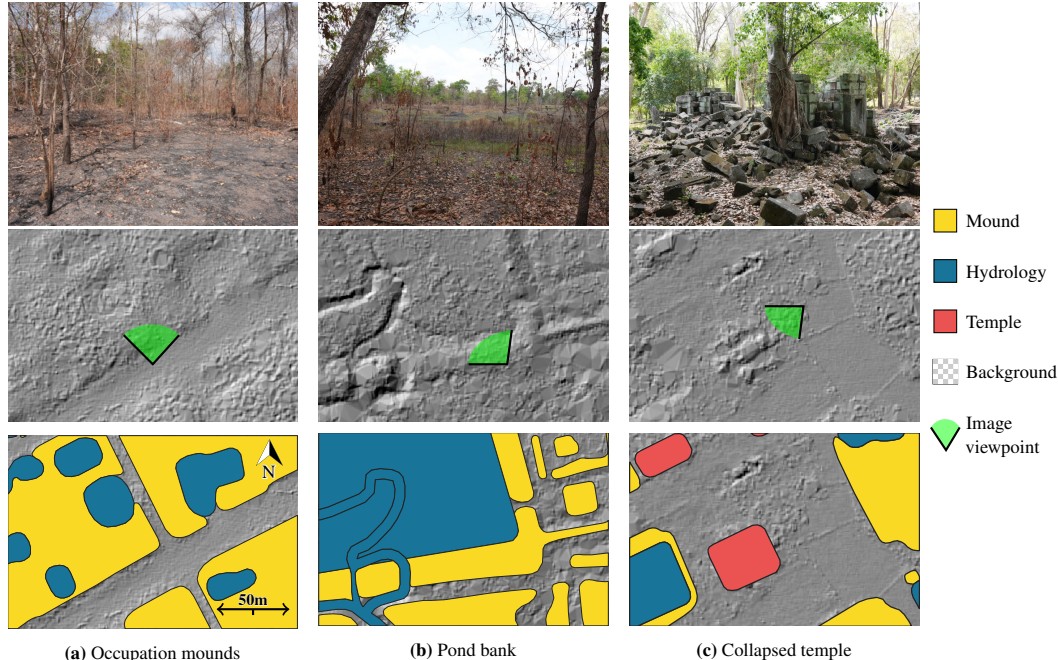

**Figure 3: Archaeoscape classes.** We illustrate the three main classes with in-situ images (top row), top-view hillshaded elevation maps (middle row), and our annotations (bottom row). In many cases, the sought features are difficult to detect visually by in-situ observation but are more apparent on elevation maps.

use cases: predicting features in a new area under a domain shift, and a realistic scenario in which archaeologists annotate part of an area of interest and train a model to pre-segment the rest.

**Misuse prevention.** There is a valid concern that large-scale annotated ALS data could be misused by malicious actors, leading to the targeted looting or destruction of historical sites [11, 12]. The potential for misuse has been a significant factor in the lack of public availability of archaeological datasets. To mitigate this risk and alleviate the concerns of local stakeholders, we propose several measures to balance the benefits of open access with the legal and practical protection of cultural heritage sites:

- **Data partitioning:** The data is divided into parcels and stripped of georeferencing and absolute elevation information to prevent spatial identification of remote, less well-known sites. While famous temples such as Angkor Wat may be recognizable, they are already under close protection by the local authorities.
- **Custom license:** The dataset is distributed under a license which forbids re-georeferencing, commercial use, and redistributing the data beyond the intended users.
- **Open credentialized access:** Access to the dataset requires signing a data agreement form, which holds users legally accountable for misuse. The Appendix contains more details about the license, data access, and the distribution agreement.

**Dataset format.** We distribute the data as GeoTIFF files with a 0.5 m resolution and polygon annotations in the GeoPackage format. We associate each pixel with the following values:

- **Radiometry:** RGB values obtained from contemporary orthophotography.
- **Ground elevation:** Digital Terrain Model (DTM) obtained with ALS, see Section 3.2.
- **Semantic label:** One-hot encoding of the five classes described below.

**Annotation.** One of the most significant undertakings of Archaeoscape is the meticulous annotation by experts, who have individually traced and field-verified a wealth of archaeological features. The annotators employed a granular classification system with 12 feature types. However, to mitigate severe class imbalance and reduce ambiguity, we have streamlined this system into a more manageable

5-class nomenclature, represented in Figure 3. We explain these classes below and provide, where applicable, the number of instances and pixel frequency:

- **Temple (827, 0.2%).** Quintessential to the Cambodian landscape, these edifices stand as the most iconic remnants of the Angkorian civilization. The scale of these temples ranges from the monumental Angkor Wat, spanning over one hundred hectares, to much smaller sites marked by little more than a scattering of bricks or stone blocks.
- **Mound (14,400, 8.6%).** Manifesting as slight elevations, these artificial earthen features are indicative of a range of human activities. They include habitation and crafting sites, as well as the embankments of canals and reservoirs. Although very numerous, mounds are often concealed by dense vegetation or too subtle to be easily detectable on the ground.
- **Hydrology (16,184, 10.4%).** This class groups together various features of Khmer hydro-engineering such as rivers, canals, reservoirs that can reach several kilometers in width, and smaller artificial ponds. These features highlight the Angkorian civilization's significant investment in water management and have long been of particular interest to archaeologists.
- **Void (3,145, 2.5%).** This annotation is reserved for areas that are considered ambiguous by expert annotators and for structures excluded from the analysis presented in this paper. Void pixels are removed from supervision and evaluation.
- **Background (78.3%).** This category encompasses everything else: regions that lack particular archaeological features or are obscured by modern development. Background includes a wide array of non-archaeological elements such as modern agricultural plots and infrastructure.

While the annotations are created and distributed as polygons, we treat them as pixel labels, framing the problem of detecting archaeological features as a conventional semantic segmentation task.

### 3.2 Acquisition and processing

**Acquisition.** ALS and orthophotography imagery was obtained during the KALC (2012) [3] and CALI (2015) [39] acquisition campaigns in Cambodia, from which a subset of 888 km$^2$ was selected, as described in Section 3.1, corresponding to over 13,000 aerial photos and 10 billion 3D points, with a density of 10-95 points per m$^2$, depending on the terrain.

The data was acquired with Leica LiDAR (ALS60 for KALC, ALS70-HP for CALI) and cameras (RCD105 and RCD30). The instruments were mounted on a pod attached to the skid of a Eurocopter AS350 B2 helicopter flying at 800 m above ground level as measured by an integrated Honeywell CUS6 IMU, and positional information was acquired by a Novatel L1/L2 GPS antenna. GPS ground support was provided by two Trimble R8 GNSS receivers.

**Preprocessing.** *Non-terrain* points (*i.e.* corresponding to tree canopies, modern buildings) are removed from ALS points with the Terrasolid software [40]. We form a DTM by fitting a triangular irregular network [41] to the remaining points and linearly interpolating the ground point elevation values within each triangular plane on a 0.5 meter grid. The photos are orthorectified and resampled to the same 0.5 meter resolution.

**Annotation.** The endeavor to map Khmer archaeological features has a long history, tracing back to the 19th century, with significant advancements following the availability of aerial imagery in the 1990s [42, 43]. Our annotation process builds upon this historical groundwork, but mostly leverages the LiDAR data collected in 2012 and 2015. Our approach relies on an iterative process of manual annotation using a Geographic Information System, QGIS, and targeted ground survey to verify features in the field. These mapping and verification efforts were performed by a shifting team of archaeologists, both local and foreign, who collectively contributed to the analysis and validation of the data. The first pre-LiDAR surveys date back to 1993, and work continued until 2024.

## 4 Benchmark

In this section, we assess the performance of modern semantic segmentation methods for ALS archaeology. We first detail how we adapt and evaluate these methods (Section 4.1), then present our results and analysis (Section 4.2), and an ablation study (Section 4.3). Finally, we discuss the limitations of our approach (Section 4.4).

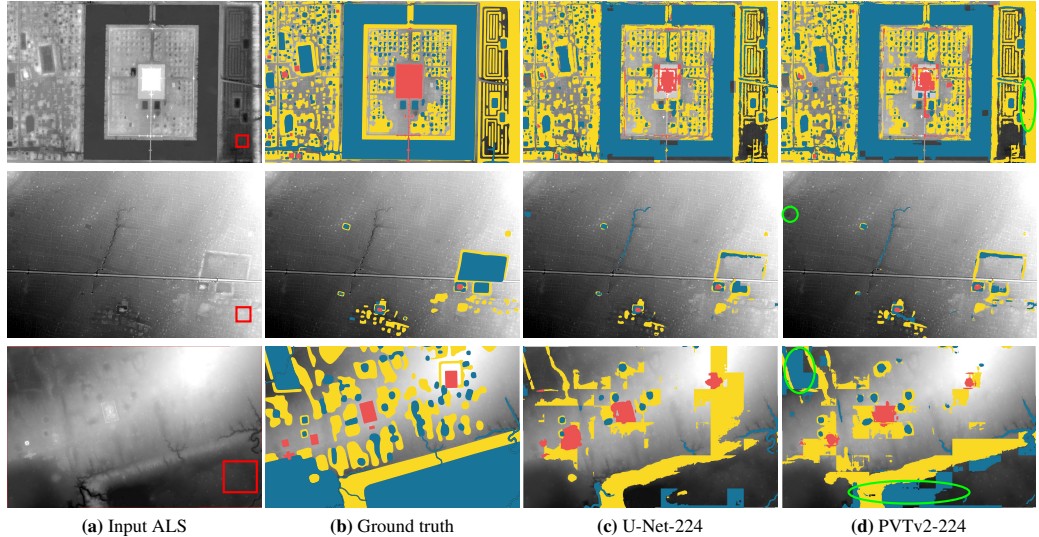

| **(a)** Input ALS | **(b)** Ground truth | **(c)** U-Net-224 | **(d)** PVTv2-224 |

**Figure 4: Qualitative performance.** We provide examples of input elevation maps (a) and their corresponding annotations (b), as well as the prediction of a standard U-Net (c) and our best model (d)—improvements in green. The red squares represent the size of the input images: 224 pixels, or 112m.

## 4.1 Baselines and metrics

We formulate the problem of finding archaeological features as a semantic segmentation task, and benchmark several backbone networks on our dataset.

**Metrics.** We evaluate the prediction of the models with the overall accuracy (OA), class-wise Intersection over Union (IoU), and the unweighted mean of the IoUs (macro-average). For the evaluation, we exclude pixels annotated with the void label.

**Implementation details.** We train the evaluated models using the configurations of the official open-source repository and provide more details in the supplementary materials. The predictions on the test set are performed along a grid corresponding to the input size and with 25% overlap on each side. Only the central portion of each prediction is kept while the border predictions are discarded.

We use a combination of internal clusters and the HPC GENCI to run our experiments. Reproducing the entire benchmark requires 260 GPU-h with A100 GPUs. We estimate the total cost of our hyperparameters search and initial experiments at 1100 GPU-h.

**Adapting baselines.** To evaluate the performance of modern vision models for ALS archaeology, we adapt several semantic segmentation models to our setting. The changes are minimal:

- **Inputs.** Beyond radiometry (RGB), we also incorporate ground elevation derived from the ALS data described in Section 3.2. As we consider networks trained on natural images, we modify the first layer to accommodate an extra band and initialize the additional weights randomly according to $\mathcal{N}(0, 0.01)$.
- **Segmentation head.** For all transformer-based methods, we map the final patch embeddings to pixel-level prediction with linear layers, except for **HybViT** which uses transposed convolutions. For CNNs, we use their dedicated segmentation heads, which we initialize randomly.
- **Pre-training and fine-tuning.** We consider models pre-trained on ImageNet1K [44] and ImageNet21K [45], but also foundation vision models trained on large external datasets: DINOv2 [36], CLIP-OPENAI [34] and LAION-2B [35], and Earth observation datasets [29, 38, 46].

## 4.2 Results

We report the quantitative performance of various state-of-the-art semantic segmentation models in Table 2, and provide qualitative examples in Figure 4.

Table 2: **Semantic segmentation benchmark.** We evaluate an array of pre-trained models fine-tuned on Archaeoscape. We first consider models with the same input size of $224 \times 224$, then present report performance for $512 \times 512$. We group models as CNNs, Vision Transformers (ViT), and hierarchical vision transformers (HViT). We **bold** the best performance for an input size of 224, and underline the performances within 0.5 point of this score. We ~frame~ the best overall performance across all resolutions.

| | Backbone | pre-training | input size | IoU avg | temple | hydro | mound | bkg | OA |
|---|---|---|---|---|---|---|---|---|---|
| CNN | U-Net[a] | ImageNet1K | 224 | 50.5 | 33.3 | 32.7 | 48.6 | 87.6 | 88.2 |
| CNN | DeepLabv3[b] | ImageNet1K | 224 | 47.6 | 19.8 | 35.9 | 47.5 | 87.2 | 87.8 |
| ViT | ViT-S[c] | ImageNet21K | 224 | 46.4 | 18.5 | 33.3 | 46.6 | 87.0 | 87.5 |
| ViT | ViT-S[c] | DINOv2 | 224 | 41.9 | 14.5 | 26.1 | 40.9 | 86.2 | 86.7 |
| ViT | ViT-B[d] | CLIP | 224 | 30.3 | 3.4 | 15.8 | 30.3 | 83.1 | 83.4 |
| ViT | ViT-B[d] | LAION2B | 224 | 32.4 | 2.8 | 14.4 | 28.2 | 84.3 | 84.6 |
| ViT | ViT-L[e] | ScaleMAE | 224 | 30.4 | 0.0 | 16.0 | 22.8 | 82.7 | 82.8 |
| ViT | HybViT-S[c] | ImageNet21K | 224 | 50.4 | 32.4 | 33.6 | 48.0 | 87.5 | 88.1 |
| ViT | DOFA[f] | DOFA | 224 | 39.6 | 13.4 | 25.9 | 33.6 | 85.5 | 86.0 |
| HViT | SWIN-S[c] | ImageNet21K | 224 | 51.9 | 33.1 | 35.2 | ~51.4~ | 88.0 | 88.6 |
| HViT | SWIN-B[g] | SatLas | 224 | 49.6 | 28.2 | 34.0 | 48.4 | 87.7 | 88.3 |
| HViT | PCPVT-S[c] | ImageNet1K | 224 | 51.7 | ~**33.4**~ | 35.0 | 50.6 | 88.0 | 88.5 |
| HViT | PVTv2-b1[c] | ImageNet1K | 224 | **52.1** | 32.3 | **36.4** | ~51.4~ | **88.2** | **88.7** |
| | U-Net[a] | ImagineNet1K | 512 | ~52.8~ | 31.8 | ~39.7~ | 50.7 | ~89.1~ | ~89.6~ |
| | PVTv2[c] | ImagineNet1K | 512 | 52.2 | 28.3 | 38.0 | 53.0 | 89.4 | 89.9 |

[a] github.com/qubvel/segmentation_models.pytorch [b] pytorch.org/vision [c] timm.fast.ai
[d] huggingface.co/laion [e] github.com/bair-climate-initiative/scale-mae
[f] https://github.com/zhu-xlab/DOFA [g] https://github.com/allenai/satlas

**Influence of the backbone.** Surprisingly, CNN-based methods such as **U-Net** outperform most ViTs on our dataset. We attribute this result to **ViT**s' reliance on extensive pre-training on RGB images. In our data, the most informative channel is the elevation rather than RGB, as the radiometric information is typically blocked by the dense canopy cover. Indeed, and as shown in Section 4.3, models trained on RGB all perform below $30\%$ mIoU. This distinction may explain why foundation models renowned for their effectiveness on natural images, such as **DINOv2** or **CLIP**, fail to adapt to this new setting. Even **ScaleMAE** and **DOFA**, which are pre-trained on large amounts of satellite imagery, lead to poor performances.

The hybrid ViT model **HybViT**, which uses convolutions for patch encoding and decoding, performs better. This suggests that integrating local feature processing (typical of CNNs) with a global perspective (a strength of ViTs) is beneficial for interpreting archaeological ALS data. This analysis is further supported by the relatively high performance of hierarchical ViT models, which even surpass CNNs in some cases. We hypothesize that the hierarchical structure of these models aligns well with the dual requirement of our task: to recognize local patterns and to integrate them within a broader spatial context. This capability is particularly advantageous for detecting archaeological features, which often consist of both small objects and expansive, interconnected structures.

**Influence of the input size.** In our experiments, we use the default size of ViTs in all experiments: 224 pixels, equivalent to 112 meters. However, the Archaeoscape dataset includes structures such as basins spanning several kilometres, and which can only be detected with a larger context. When scaling our input size to 512, we noted a significant improvement in performance, especially with the **U-Net** model. Attempts to further increase the input size did not yield additional performance gains, as the models quickly overfit to the training set.

**Qualitative analysis.** As depicted in the top row of Figure 4, models trained on our data can detect complex structures, such as the central grid inside the temple moat and the maze-like features outside. However, they miss the broader semantic context, *e.g.* finding the prominent temple walls while

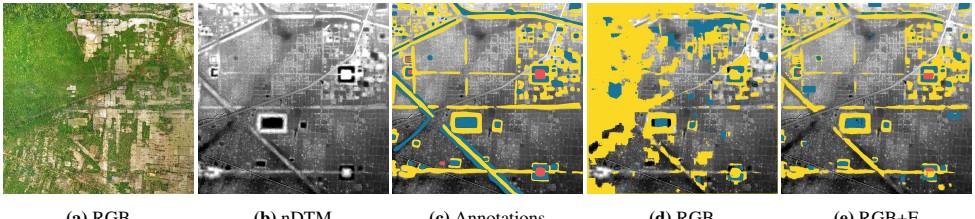

| **(a)** RGB | **(b)** nDTM | **(c)** Annotations | **(d)** RGB | **(e)** RGB+E |

**Figure 5: Channel ablation.** We represent the orthophotography (a), normalized terrain model (b), and annotations (c). We also provide the prediction of a PVTv2 model operating on RGB photos (d), and a model processing both RGB and elevation data (e). The model using only radiometric information performs worse overall, and in particular, fails to identify any structures under the heavily forested area at the top left corner.

failing to segment the platform. In the middle row, the models detect isolated features and temples with the standard "horseshoe" configuration, while the large ponds are mostly missed, likely due to the limited context window size. In the bottom row, the hilly areas with faint feature elevation pose a significant challenge. This highlights the limitations of current models in handling the varying landscape, scale, and semantic context of archaeological features.

**Overall performance.** The performance across models remains relatively low, especially if compared to results achieved on complex computer vision segmentation benchmarks featuring numerous classes. This suggests that contemporary model architectures may not adequately meet the specific challenges of ALS archaeology, which involves interpreting subtle local elevation patterns within a broader spatial context. Furthermore, foundation models for natural images often fail to adapt to the specificity of the data and the new elevation channel, possibly due to their extensive pre-training. This situation highlights the need for bespoke models specifically tailored for ALS data analysis.

### 4.3 Ablation study

We evaluate the impact of some of the choices made in the design of Archaeoscape through an ablation study.

**Channel importance.** Airborne LiDAR scans are pivotal for uncovering the subtle elevation patterns of archaeological features like mounds and canals, which are typically not visible in orthophotos, as shown in Figure 5. Moreover, dense canopies can obscure or completely hide radiometric information about the ground. Conversely, in less densely forested areas, orthophotos can capture detailed information about archaeological features, complementing LiDAR data. The ablation study results, documented in Table 4, highlight the limitations of relying solely on RGB data. Models using only RGB information registered a mean Intersection over Union (mIoU) of about 30%, significantly lower than models also utilizing elevation data. This disparity underscores the inadequacy of RGB data under dense canopy coverage. Furthermore, while removing RGB information only moderately affects performance, it particularly affects the detection of temples——some of which are still standing to this day, and are typically not covered by the canopy. The performance gap between models pretrained with DINOv2 and those pretrained on ImageNet widens without RGB, suggesting that DINOv2 models are highly optimized for RGB processing, whereas ImageNet models adapt better to elevation data.

**Initialization strategy.** Adapting models trained on RGB data to handle elevation channels poses challenges. Our approach, detailed in Section 4, initialize with small values the weights of the first layer corresponding to the new channel while retaining the pre-trained weights for RGB. In Table 4, we evaluate this method against three alternatives: fully random initialization, random initialization of the first layer with other weights retained, and LoRA fine-tuning. Randomly initializing the first layer results in performance akin to training the network from scratch, demonstrating the efficacy of our strategy to leverage pre-existing RGB training.

### 4.4 Limitations

Archaeoscape presents several limitations as a benchmark that should be considered:

**Table 4: Ablation study.** We evaluate the impact of omitting RGB channels or elevation from input images and assess various initialization strategies for fine-tuning networks initially trained only on RGB data to accommodate additional elevation channels E. Performance metrics are highlighted, with the best scores bolded and those within 0.5 points underlined.

| | | | | IoU | | | | OA |
|---|---|---|---|---|---|---|---|---|
| | | | **avg** | temple | hydro | mound | bkg | |
| Channels | backbone | pretraining | | | | | | |
| RGB+E | U-Net | ImageNet1K | 50.5 | **33.3** | 32.7 | 48.6 | 87.6 | 88.2 |
| | ViT-S | DINOv2 | 41.9 | 14.5 | 26.1 | 40.9 | 86.2 | 86.7 |
| | PVTv2-b1 | ImageNet1K | **52.1** | 32.3 | 36.4 | **51.4** | 88.2 | 88.7 |
| E | U-Net | ImageNet1K | 51.2 | 28.8 | **37.8** | 49.8 | **88.3** | **88.9** |
| | ViT-S | DINOv2 | 36.6 | 10.4 | 19.4 | 31.5 | 85.2 | 85.6 |
| | PVTv2-b1 | ImageNet1K | 49.9 | 27.8 | 35.1 | 48.5 | 88.0 | 88.5 |
| RGB | U-Net | ImageNet1K | 34.2 | 1.5 | 22.7 | 29.3 | 83.2 | 83.2 |
| | ViT-S | DINOv2 | 29.0 | 1.6 | 12.6 | 20.1 | 81.6 | 81.4 |
| | PVTv2-b1 | ImageNet1K | 33.9 | 6.0 | 20.6 | 27.0 | 82.0 | 33.9 |
| Initialization | backbone | pretraining | | | | | | |
| Fully random | | | 46.0 | 17.9 | 32.7 | 46.1 | 87.2 | 87.7 |
| Rand. 1st layer | PVTv2-b1 | ImageNet1K | 44.4 | 17.5 | 28.2 | 46.1 | 87.2 | 86.5 |
| LoRA (rank 32) | | | 46.1 | 21.8 | 31.9 | 44.4 | 86.2 | 86.6 |
| Proposed | | | **52.1** | **32.3** | **36.4** | **51.4** | **88.2** | **88.7** |

- **Domain shift:** The imagery for Archaeoscape has been collected over two campaigns using different equipment. Even with our best efforts to harmonize the dataset and its processing, sensor and meteorological variations may manifest in the data distribution.
- **Annotation errors and ambiguity:** As they were annotated and field-verified by expert archaeologists, we can affirm that the annotated polygons correspond to actual archaeological features with high confidence. However, there is an inevitable degree of ambiguity regarding the precise shape and boundaries of these features, which often consist of very slight relief sloping gradually into the natural terrain. Moreover, we cannot rule out that background terrain may contain some yet uncovered features that would have eluded detection.
- **Cultural specificity:** Archaeoscape aims to serve as a benchmark for vision models for ALS archaeology, but focuses exclusively on the Khmer civilization. We acknowledge that our conclusions may not be universally applicable to other cultural contexts or regions.

# 5    Conclusion

We have introduced Archaeoscape, the largest published dataset for ALS archaeology featuring open-access imagery and annotations. Focused on the ancient Khmer settlement complexes and temples of Cambodia, our dataset covers $888$ km$^2$ and comprises $31,144$ individual anthropogenic instances. We provide an extensive benchmark evaluating several state-of-the-art computer vision models for detecting archaeological features within elevation maps and images. Despite formulating the problem as a classic semantic segmentation task, we observe that even usually high-performing models struggle to achieve high scores. We attribute this poor performance to the unique challenges of ALS archaeology, such as the subtlety of the patterns sought, and the importance of large-scale context. We hope that our dataset will encourage the computer vision and machine learning community to propose novel solutions for these unresolved challenges.

## Acknowledgments and disclosure of funding

The experiments conducted in this study were performed using HPC/AI resources provided by GENCI-IDRIS (Grant 2023-AD011014781).

This work has made use of results obtained with the Chalawan HPC cluster, operated and maintained by the National Astronomical Research Institute of Thailand (NARIT) under the Ministry of Science and Technology of Royal Thai government.

This project is funded by the European Research Council (ERC) under the European Union's Horizon 2020 research and inovation programme (grant agreement No 866454).

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

# APPENDIX

## A.1 Checklist


## A.2 Author statement

The authors hereby acknowledge and accept full responsibility for the content of the Archaeoscape dataset. They confirm that this dataset does not violate any intellectual property rights, privacy rights, or other legal or ethical standards.

They indemnify and hold harmless the NeurIPS Dataset and Benchmark Track from any claims, damages, or legal actions resulting from the submission, storage, or use of this dataset.

## A.3 Additional ALS archaeology-related datasets

ALS data are well-used by archaeologists for its precision and ability to recover the archaeological features [10]. Several recent works have leveraged deep learning techniques to automatically detect features of interest. In Table A.1, we provide a list of such works. Note that this is a very dynamic field, and this list may not be exhaustive.

| | open-access | hi-res RGB | location | extent in km$^2$ | resolution in meters | number of instances |
|---|---|---|---|---|---|---|
| AHN [47] | ✗ | ✗ | Netherlands | 437 | 0.5 | N/A |
| Mysteries of the Maya [48] | ✗ | ✗* | Mexico | 120 | 0.5 | N/A |
| Jönköping [49] | ✗ | ✗ | Sweden | 22 | 1 | 155 |
| Mégalithes de Bretagne [5] | ✗ | ✗ | France | 200 | 0.5 | 195 |
| Białowieża Forest [6] | ✗ | ✗ | Poland | N/A | 0.5 | 211 |
| Galicia [50] | ✗ | ✗* | Spain | N/A | 1 | 306 |
| Arran [13] | ✔ | ✗ | United Kingdom | 25 | 0.5 | 772 |
| Sápmi [51] | ✗ | ✔ | Finland | 21 | N/A | 997 |
| MayaArch3D [52] | ✗ | ✗ | Honduras | 25 | N/A | 1,124 |
| Litchfield [14] | ✗ | ✗ | USA | 50 | 1 | 1,866 |
| Puuc [15] | ✗ | ✗ | Mexico | 22.5 | 0.5 | 1,966 |
| AHN [16] | ✗ | ✗ | Netherlands | 81 | 0.5 | 3,553 |
| AHN-2 [17] | ✗ | ✗ | Netherlands | 437 | 0.5 | 3,849 |
| Connecticut [18] | ✗ | ✗ | USA | 353 | 1 | 3,881 |
| Pennsylvania [20] | ✔ | ✗ | USA | 4 | 1 | 4,376 |
| Dartmoor [19] | ✗ | ✗ | United Kingdom | 12 | 0.5 | 4,726 |
| Uaxactun [21] | ✗ | ✗ | Guatemala | 160 | 1 | 5,080 |
| Lusatia [53] | ✗ | ✗ | Germany | 3.4 | 0.5 | 6,000 |
| Chactún [22] | ✗ | ✗ | Mexico | 230 | 0.5 | 10,894 |
| **Archaeoscape (ours)** | ✔ | ✔ | Cambodia | **888** | 0.5 | **31,411** |

Table A.1: **ALS archaeology datasets.** We list ALS datasets used for archeological analysis. N/A stands for information we could not find. ✶ Sentinel-1 and 2 imagery are provided.

## A.4 Additional results

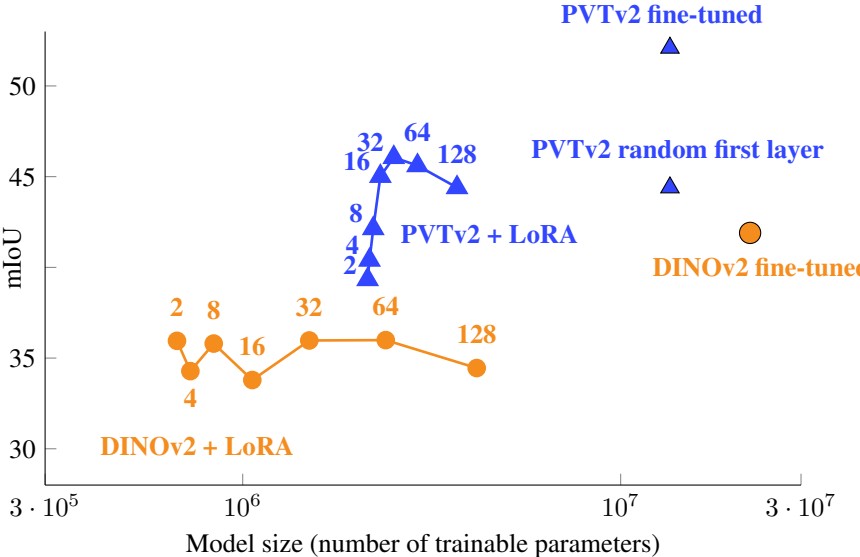

**Figure A.1: DINOv2 fine-tuning vs LoRA.** We fine-tuned DINOv2 small with LoRA at different ranks and compared their mIoU. Lower ranks perform better but are still 5% behind full fine-tuning.

Foundation models can be difficult to fine-tune to new datasets or tasks, as they tend to easily overfit. Low rank adaptation (LoRA) [54] can remedy this issue by only learning a low-rank update to the weights of the pre-trained model. Figure Figure A.1 provides the performance of PVTv2 and DINO models fine-tuned using LoRA compared to a fully fine-tuned.

When fine-tuned with LoRA, the performance of DINO rapidly plateaus and even decreases, showing that the learned features can not be easily adapted from RGB images to terrain models. Conversely, the performance of PVTv2 increases with the rank used for LoRA, but does not reach the performance of a fully fine-tuned network. This suggests that, when fine-tuned to new input and target domains, and particularly when using LoRA, large models can become over-adapted to their source domain and struggle to generalize.

We provide additional visualizations of the mapping outputs generated by our models in Figure A.2. Those once again illustrate the difficulty that arises from large-scale dependency, particularly in water prediction. Columns 2 and 3 are respectively an illustrations of a failure and success case in predicting large bodies of water. While our model is able to accurately identify most temples and mounds, the reconstruction of the exact shape of religious or settlement complexes remains approximate. Moreover, we observe that the model struggles to detect fainter mounds, although these are still visible to human experts.

## A.5 Implementation details

### A.5.1 Data split

We designed the split to each contain emblematic archaeological features—large-scale hydraulic engineering sites, monumental temples, subtle features, and typical terrain types—dense and scarce occupation, hills, and floodplains. The class distribution per split is given in Table A.2.

### A.5.2 Data loader

Our data loader loads images of size $224 \times 224$ pixels at a resolution of $50$ cm, with RGB channels normalized using the dataset mean and variance. The elevation channel is normalized separately,

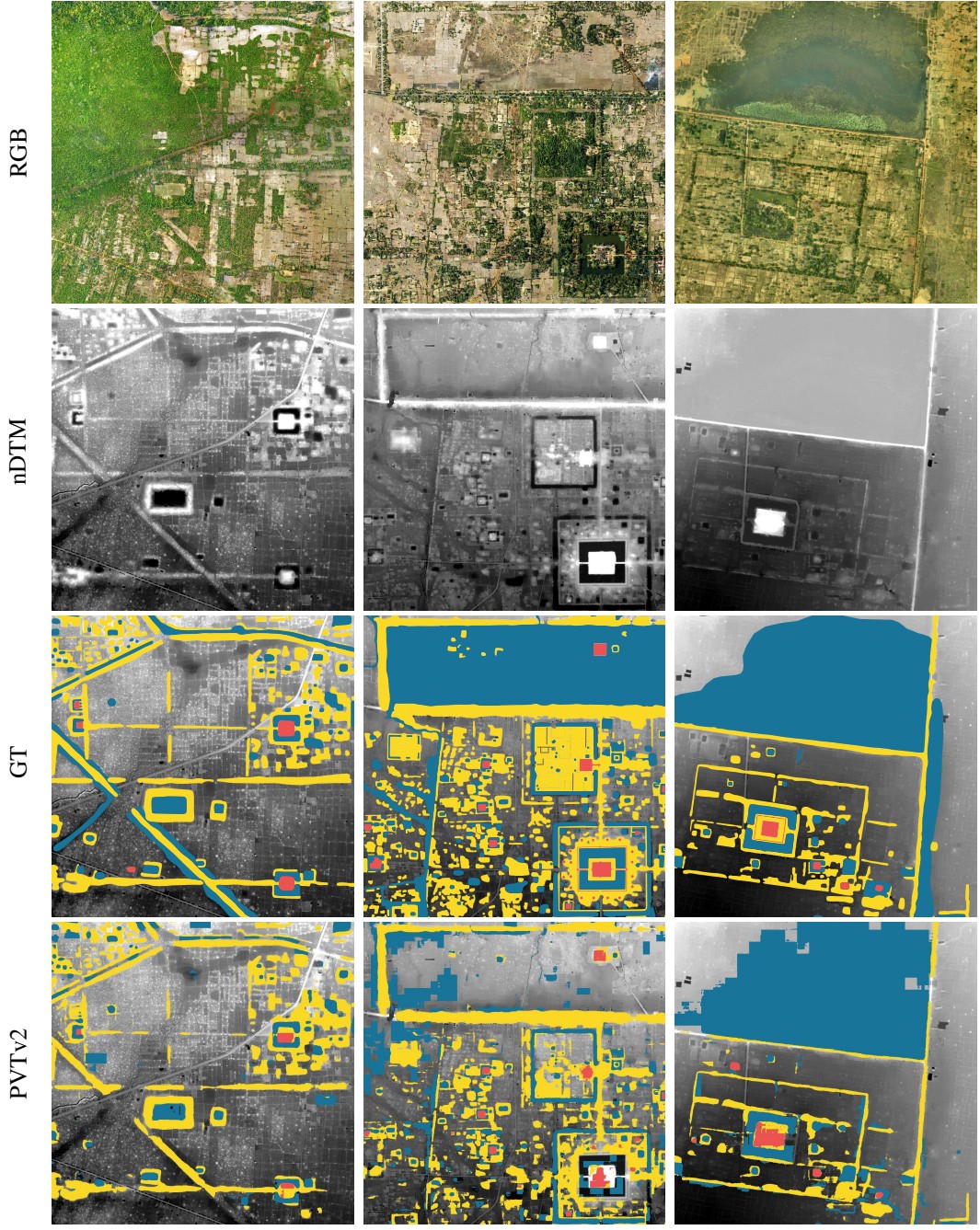

**Figure A.2: Qualitative illustrations.** We present predictions from our best model (PVTv2) on three challenging areas. For each region, we give the RGB ortho-photo (RGB), the normalized elevation map (nDTM), the ground truth (GT) as well as the model prediction (PVTv2).

using the mean and variance of each sampled image. We use a batch size of 64 throughout all experiments.

To sample random images during training, we first randomly select pixels from one of the predefined study areas with a probability proportional to their area. We then take a crop of size $224 \times 224$ centered on this pixel. The image is rejected if over $80\%$ of its extent falls out of the area. otherwise, the out-of-area pixels are padded with the image's mean for each channel. Finally, we apply the following augmentations, each activated with an independent probability of $0.5$:

**Table A.2: Class distribution per split.**

| Split | Temple | Hydro | Mound | Background |
|---|---|---|---|---|
| Overall | 0.25% | 10.7% | 8.9% | 80.2% |
| Train | 0.23% | 9.7% | 8.4% | 81.7% |
| Val | 0.25% | 20.6% | 10.8% | 68.4% |
| Test | 0.34% | 8.9% | 9.5% | 81.2% |

- **Scale modification:** The image is scaled by a random factor between $0.5$ and $2$.
- **Rotation**: The sample is rotated by an angle randomly chosen between $-90°$ and $90°$ using bilinear interpolation.

During validation, we sample pixels along a regular grid with a $224$ pixels step. During testing, we use a $112$ pixels grid, and, when predicted images overlap, only keep the center half of the image, *i.e.* discarding the 25% border on all sides. We do not employ augmentations during the evaluation.

### A.5.3 Training

We use the ADAM optimizer with a linear warm-up schedule that increases the learning rate from $10^{-5}$ to $10^{-3}$ across the first two epochs of training. We use a `ReduceLROnPlateau` [55] learning rate scheduler with a patience of $4$ and a decay of $5$.

## A.6 License

The Archaeoscape dataset is under a custom license, which prevents redistribution and attempts at localizing the data. We provide the full text of the license below.

*The École française d'Extrême-Orient (EFEO) makes the Archaeoscape dataset (the "DATASET") available for research and educational purposes to individuals or entities ("USER") that agree to the terms and conditions stated in this License.*

1. *The USER may access, view, and use the DATASET without charge for lawful non-commercial research purposes only. Any commercial use, sale, or other monetization is prohibited. The USER may not use the DATASET for any unlawful activities, including but not limited to looting, vandalism, and disturbance of archaeological sites.*
2. *The USER may not attempt to identify the location of any part of the DATASET and must exercise all reasonable and prudent care to avoid the disclosure of the locations referenced in the DATASET in any publication or other communication.*
3. *The USER may not share access to the DATASET with anyone else. This includes distributing the download link or any portion of the DATASET. Other users must register separately and comply with all the terms of this Licence.*
4. *The USER must use the DATASET in a manner that respects the cultural heritage of Cambodia and its people, and in compliance with the relevant Cambodian authorities. Any use of the DATASET that could harm or exploit these cultural sites or their environment is strictly prohibited.*
5. *The USER must properly attribute the EFEO as the source of the data in any publications, presentations, or other forms of dissemination that make use of the DATASET.*
6. *This agreement may be terminated by either party at any time, but the USER's obligations with respect to the DATASET shall continue after termination. If the USER fails to comply with any of the above terms and conditions, their rights under this License shall terminate automatically and without notice.*

*THE DATASET IS PROVIDED "AS IS," AND THE EFEO DOES NOT MAKE ANY WARRANTY OF ANY KIND, EXPRESS OR IMPLIED, INCLUDING BUT NOT LIMITED TO WARRANTIES OF MERCHANTABILITY, FITNESS FOR A PARTICULAR PURPOSE, AND NON-INFRINGEMENT. IN NO EVENT SHALL THE EFEO OR ITS COLLABORATORS BE LIABLE FOR ANY CLAIM, DAMAGES, OR OTHER LIABILITY ARISING FROM THE USE OF THE DATASET.*

## A.7 Datasheet for dataset

### A.7.1 Motivation

Q1 **For what purpose was the dataset created?** Was there a specific task in mind? Was there a particular gap that needed to be filled? Please provide a description.

- The Archaeoscape dataset is an open-access ALS dataset intended for archaeology. It is simultaneously the largest in terms of its extent and number of annotated anthropogenic features. The intended task is the semantic segmentation of LiDAR-derived terrain maps to find archaeological traces and structures under dense vegetation.

Q2 **Who created the dataset (e.g., which team, research group) and on behalf of which entity (e.g., company, institution, organization)?**

- The different parts of the dataset were acquired during several acquisition campaigns as part of the Khmer Archaeology Lidar Consortium (KALC) and Cambodian Archaeological Lidar Initiative (CALI), joint programs of which the EFEO (Ecole française d'Extrême-Orient) was a member. The curation and benchmarking were performed jointly with the IMAGINE team (A3SI/LIGM, ENPC).

Q3 **Who funded the creation of the dataset?** If there is an associated grant, please provide the name of the grantor and the grant name and number.

- The ALS acquisitions were funded by the following parties:
    - European Research Council (ERC)
    - École française d'Extrême-Orient (EFEO)
    - University of Sydney (USYD)
    - Société Concessionnaire d'Aéroport (SCA/INRAP Airport)
    - Hungarian Indochina Company (HUNINCO)
    - Japan-APSARA Safeguarding Angkor (JASA)
    - Archaeology & Development Foundation Phnom Kulen Program (ADF Kulen)
    - World Monuments Fund (WMF)
- And are associated with the following ERC grants:
    - CALI: "The Cambodian Archaeological Lidar Initiative: Exploring Resilience in the Engineered Landscapes of Early SE Asia" (Grant agreement ID: 639828)
    - archaeoscape.ai: "Exploring complexity in the archaeological landscapes of monsoon Asia using lidar and deep learning" (Grant agreement ID: 866454).
- The funding for the Archaeoscape annotations is 100% public. The EFEO is an "*Établissement public à caractère scientifique, culturel et professionnel*", *i.e.* a public scientific, cultural or professional establishment which is financed by public funds.

Q4 **Any other comments?**

- [N/A]

### A.7.2 Composition

Q5 **What do the instances that comprise the dataset represent (e.g., documents, photos, people, countries)?**

- The dataset covers several sites in Cambodia of archaeological interest. The dataset comprises ALS-derived elevation maps, orthorectified photography, and manually annotated archaeological features.

Q6 **How many instances are there in total (of each type, if appropriate)?**

- Archaeoscape covers 888 km$^2$ and 31,141 individual archaeological features. The dataset is split into 23 non-overlapping parcels, from 2 to 183 km$^2$.

Q7 **Does the dataset contain all possible instances or is it a sample (not necessarily random) of instances from a larger set?**

- Archaeoscape covers only a fraction of the full extent of the Khmer Empire at its apogee and of the likely distribution of Khmer archaeological features in the landscape. Those parts of the dataset contained in the training, validation, and test sets have been

extensively annotated by archaeological experts. We can affirm with a reasonably degree of confidence that the vast majority of the archaeological features in these splits have been identified and annotated.

Q8 **What data does each instance consist of?**

- Each parcel is a raster file under the GeoTIFF format with a ground sampling distance of $0.5$ m. Each pixel is associated with: (i) a terrain elevation relative to the lowest point of the file, (ii) an RGB value derived from an orthorectified aerial photograph, (iii) where available, a label corresponding to one of the sought classes, (iv) a binary value indicating whether or not the pixel is in the parcel.

Q9 **Is there a label or target associated with each instance?**

- [Yes] We provide dense pixel-precise annotations for $888$ km$^2$ corresponding to over 3.5 billion annotated pixels.

Q10 **Is any information missing from individual instances?**

- [Yes] The georeferencing information has been stripped from the dataset parcels.

Q11 **Are relationships between individual instances made explicit (e.g., users' movie ratings, social network links)?**

- [No] To prevent their re-georeferencing, we have purposefully removed any information on the relationships between parcels.

Q12 **Are there recommended data splits (e.g., training, development/validation, testing)?**

- [Yes] We provide the following data splits: train, validation and test. The test split has been explicitly selected to contain a representative variety of configurations. We implement a 100 m buffer between all parcels.

Q13 **Are there any errors, sources of noise, or redundancies in the dataset?**

- As the annotations are made through visual interpretation with quality control, some errors are unavoidable, especially for classes that are visually hard to distinguish. Some unavoidable noise occurs due to the ambiguous boundaries of subtle archaeological features. Internal quality control has been performed to limit such errors. There are no redundancies in the dataset, each parcel covers a distinct area.

Q14 **Is the dataset self-contained, or does it link to or otherwise rely on external resources (e.g., websites, tweets, other datasets)?**

- This dataset is self-contained and will be stored and distributed by the EFEO.

Q15 **Does the dataset contain data that might be considered confidential (e.g., data that is protected by legal privilege or by doctor–patient confidentiality, data that includes the content of individuals' non-public communications)?**

- [No] The data does not contain confidential information. However, to limit potential misuse such as looting or destruction of historical sites, the georeferencing and absolute elevation of the parcels have been removed.

Q16 **Does the dataset contain data that, if viewed directly, might be offensive, insulting, threatening, or might otherwise cause anxiety?** *If so, please describe why.*

- [No]

Q17 **Does the dataset identify any subpopulations (e.g., by age, gender)?**

- [No]

Q18 **Is it possible to identify individuals (i.e., one or more natural persons), either directly or indirectly (i.e., in combination with other data) from the dataset?**

- [No] The nDTM elevation data excludes extraneous points such as modern buildings. The RGB orthophotography resolution of 50 cm/pixel and the aerial perspective prevent the recognition of individuals.

Q19 **Does the dataset contain data that might be considered sensitive in any way (e.g., data that reveals racial or ethnic origins, sexual orientations, religious beliefs, political opinions or union memberships, or locations; financial or health data; biometric or genetic data; forms of government identification, such as social security numbers; criminal history)?**

- [No]

Q20 **Any other comments?**

- The safe and ethical release of archaeological data has been the subject of numerous studies [12]. We have implemented the best practices of the field to minimize potential risks of misuse.

### A.7.3 Collection Process

Q21 **How was the data associated with each instance acquired?**

- The ALS data and photography were acquired from aerial surveys in Cambodia and mapped onto a cartographic coordinate reference system. From this data a subset of 888 km$^2$ was selected, corresponding to over 13,000 aerial photos and 10 billion points, with a density of 10-95 points per m$^2$, depending on the terrain.
- The ALS points were filtered to remove noise and classified. The normalized Digital Terrain Models (nDTM) (relative ground elevation) was obtained from the classified ALS point clouds using open-source software. A triangular irregular network was fitted to the *ground* points (excluding extraneous elements such as tree canopies and modern buildings), with a DTM formed by linear interpolation of the elevation values within each triangular plane based on a 0.5 meter grid. The same procedure was applied to obtain intensity and return number metadata maps. The photos were orthorectified and resampled to the same 0.5 meter resolution.

Q22 **What mechanisms or procedures were used to collect the data (e.g., hardware apparatus or sensor, manual human curation, software program, software API)?**

- The data was acquired with Leica LiDAR (ALS60 for KALC, ALS70-HP for CALI) and cameras (RCD105 and RCD30). The instruments were mounted on a pod attached to the skid of a Eurocopter AS350 B2 helicopter flying at 800 m above ground level as measured by an integrated Honeywell CUS6 IMU, and positional information acquired by a Novatel L1/L2 GPS antenna. GPS ground support was provided by two Trimble R8 GNSS receivers.

Q23 **If the dataset is a sample from a larger set, what was the sampling strategy (e.g., deterministic, probabilistic with specific sampling probabilities)?**

- The target areas for the LiDAR acquisition campaigns were selected on the grounds of archaeological value and interest by domain experts. A subset of 888 km$^2$ presented in this dataset was selected by choosing 23 non-overlapping parcels in the areas where archaeological annotations were deemed complete and finalized, preserving the global distribution of features and landscapes across the training, validation and test sets.

Q24 **Who was involved in the data collection process (e.g., students, crowdworkers, contractors) and how were they compensated (e.g., how much were crowdworkers paid)?**

- These mapping and verification efforts were performed by a shifting team of archaeologists, both local and foreign, who collectively contributed to the analysis and validation of the data, with the first pre-LiDAR surveys dating back to 1993, and continuing until 2024. All persons involved were employees and researchers from foreign governmental institutions, such as the EFEO or Sydney University, or employed by the Cambodian governmental authorities, following strictly existing ethical codes and national regulations.

Q25 **Over what timeframe was the data collected? Does this timeframe match the creation timeframe of the data associated with the instances (e.g., recent crawl of old news articles)?**

- The KALC campaign took place in 2012, and the CALI campaign in 2015. The annotations are the result of a continuous effort from 1993 to 2024.

**Q26 Were any ethical review processes conducted (e.g., by an institutional review board)?**

- [Yes]  Yes, as a part of the CALI and archaeoscape.ai ERC grants.

**Q27 Does the dataset relate to people?**

- The dataset describes the archaeological remains of anthropogenic structures, but does not directly relate to living people.

**Q28 Did you collect the data from the individuals in question directly, or obtain it via third parties or other sources (e.g., websites)?**

- [N/A]

**Q29 Were the individuals in question notified about the data collection?**

- [N/A]

**Q30 Did the individuals in question consent to the collection and use of their data?**

- [N/A]

**Q31 If consent was obtained, were the consenting individuals provided with a mechanism to revoke their consent in the future or for certain uses?**

- [N/A]

**Q32 Has an analysis of the potential impact of the dataset and its use on data subjects (e.g., a data protection impact analysis) been conducted?**

- [Yes]  We have studied potential misuse of the data and have taken steps to prevent it, such as removing and obfuscating the location of acquisitions.

**Q33 Any other comments?**

- [No]

### A.7.4   Preprocessing, cleaning, and/or labeling

**Q34 Was any preprocessing/cleaning/labeling of the data done (e.g., discretization or bucketing, tokenization, part-of-speech tagging, SIFT feature extraction, removal of instances, processing of missing values)?**

- [Yes]  The ALS acquisitions are delivered in the form of 3D point clouds. The ALS points were filtered to remove noise and classified. We have extracted terrain models from the *ground* clouds only, *i.e.* those not belonging to the tree canopies and modern buildings.

**Q35 Was the "raw" data saved in addition to the preprocessed/cleaned/labeled data (e.g., to support unanticipated future uses)?** *If so, please provide a link or other access point to the "raw" data.*

- [Yes]  The data has been saved, but will not be distributed to prevent data re-localization.

**Q36 Is the software used to preprocess/clean/label the instances available?**

- [Yes]  The annotation software is open source. The anonymized data preprocessing code (without references to specific locations or coordinates) is available.

**Q37 Any other comments?**

- [No]

### A.7.5   Uses

**Q38 Has the dataset been used for any tasks already?**

- [Yes]  As part of the KALC and CALI projects, and for archaeological publications, but not in an open-access fashion.

**Q39 Is there a repository that links to any or all papers or systems that use the dataset?**

- [Yes]  Such a list will be made available on the website of the project.

**Q40 What (other) tasks could the dataset be used for?**

- Beyond its use as a difficult semantic segmentation benchmark, the Archaeoscape data holds significant value for archaeologists of the Khmer cultural world who seek to employ machine learning models for feature annotations, and possibly for a wider archaeological audience as well.

Q41 **Is there anything about the composition of the dataset or the way it was collected and preprocessed/cleaned/labeled that might impact future uses?**

- As the localization of the acquisitions has been removed and obfuscated, the direct archaeological utility of the dataset in its present form is **necessarily** limited.
- By choosing a subset of 23 non-overlapping parcels covering 888 km$^2$, we have removed the spatial and cultural relations between them, which will be a concern for researchers seeking to incorporate that information.

Q42 **Are there tasks for which the dataset should not be used?**

- [Yes] Attempting to perform registration and re-localization of the dataset is explicitly forbidden by the dataset license, as well as all commercial use.

Q43 **Any other comments?**

- [No]

### A.7.6 Distribution

Q44 **Will the dataset be distributed to third parties outside of the entity (e.g., company, institution, organization) on behalf of which the dataset was created?**

- [Yes] The dataset will be limited to users who agree to its licence and can provide access credentials, as part of the credentialized open access distribution policy.

Q45 **How will the dataset be distributed (e.g., tarball on website, API, GitHub)?**

- The data will be available through a web-platform maintained by the EFEO.

Q46 **When will the dataset be distributed?**

- The dataset will be distributed upon acceptance of this paper, and will be made public at the camera-ready deadline at the latest.

Q47 **Will the dataset be distributed under a copyright or other intellectual property (IP) license, and/or under applicable terms of use (ToU)?** *If so, please describe this license and/or ToU, and provide a link or other access point to, or otherwise reproduce, any relevant licensing terms or ToU, as well as any fees associated with these restrictions.*

- [Yes] The dataset will be released under a custom license which forbids its distribution and attempts to localize the data.

Q48 **Have any third parties imposed IP-based or other restrictions on the data associated with the instances?**

- [No]

Q49 **Do any export controls or other regulatory restrictions apply to the dataset or to individual instances?**

- [No]

Q50 **Any other comments?**

- [No]

### A.7.7 Maintenance

Q51 **Who will be supporting/hosting/maintaining the dataset?**

- The EFEO will support and host the dataset and its metadata.

Q52 **How can the owner/curator/manager of the dataset be contacted (e.g., email address)?**

- archaeoscape@efeo.net
- christophe.pottier@efeo.net

Q53 **Is there an erratum?**

- [No] There is no erratum for our initial release. Errata will be documented as future releases on the dataset website.

Q54 **Will the dataset be updated (e.g., to correct labeling errors, add new instances, delete instances)?**

- We do not plan to update this dataset, but may release an expansion in the future.

Q55 **If the dataset relates to people, are there applicable limits on the retention of the data associated with the instances (e.g., were individuals in question told that their data would be retained for a fixed period of time and then deleted)?**

- [N/A]

Q56 **Will older versions of the dataset continue to be supported/hosted/maintained?**

- [Yes] The EFEO is dedicated to providing ongoing support for the Archaeoscape dataset.

Q57 **If others want to extend/augment/build on/contribute to the dataset, is there a mechanism for them to do so?**

- [No] Our license forbids third-party redistribution of any portion of the dataset.

Q58 **Any other comments?**

- [No]

