# OpenReview forum: "Archaeoscape: Bringing Aerial Laser Scanning Archaeology to the Deep Learning Era"
_NeurIPS.cc/2024/Datasets_and_Benchmarks_Track — NeurIPS 2024 Track Datasets and Benchmarks Spotlight_

### Official Review · Reviewer_AJFq · 2024-07-07
**A dataset for archaeology**

**Rating:** 6
**Confidence:** 4
**Correctness:** Yes
**Clarity:** Yes

**Review:**

please see Strengths and Limitations

**Strengths:**

The article has two contributions:
1. Dataset Construction: An ALS dataset covering an area of 888 square kilometers was constructed, containing 31,411 instances of archaeological features. Detailed manual annotations and field verifications were conducted.

2. Model Evaluation: Multiple modern semantic segmentation models (such as U-Net, DeepLabv3, Vision Transformers, etc.) were benchmarked, and the performance of different models on this dataset was compared.

**Additional Feedback:**

It is hoped to conduct the expected experiments.

**Documentation:**

The dataset visit link is provided.

**Limitations:**

1. In the analysis of the Influence of the backbone in section 4.2, the authors believe that integrating local feature processing with a global perspective is beneficial for interpreting archaeological ALS data. They attribute the effectiveness of methods like HybViT to this. However, ScaleMAE also integrates local feature processing with a global perspective, which is contradictory. The authors should further investigate the reasons behind the performance advantages of methods like HybViT.

2. The authors simply hypothesize that these models align well with the dual requirement of recognizing local patterns and integrating them within a broader spatial context. However, in specific remote sensing fields, there are various pre-trained foundational models that adapt to this local-global task, including ScaleMAE and so on. Since the authors' work involves a specific dataset, they should focus on benchmarking their performance against various vertical domain tasks that are relevant, not just general tasks. They should conduct more comprehensive research and experiments on pre-training work and segmentation work in their specific or related fields.

3. The proposed dataset includes different channels, and the authors conducted ablation experiments on these channels. The dataset comprises LiDAR and RGB data, but the authors did not further consider how to integrate these two data sources. In the experiments showcasing the combined effect of these two data sources, how were they aligned? Furthermore, the authors need to explore whether the information from these two data sources in the dataset complements each other and whether they can be jointly used to improve performance. Simply using a backbone seems insufficient; should the approach be expanded to joint processing of multi-source data? The authors should also explain in detail how the experimental results were achieved using the two data sources.

**Opportunities For Improvement:**

Please address the three issues raised in the limitations.

**Relation To Prior Work:**

Yes

**Summary And Contributions:**

The article introduces Archaeoscape, a large-scale open-source dataset designed to leverage deep learning techniques for analyzing airborne laser scanning (ALS) data and addressing challenges in traditional archaeological methods.
While the Archaeoscape dataset holds significant potential for advancing archaeology, the article's benchmarking and experimental evaluations are insufficient and need more thorough testing.

---

> ### Author Rebuttal · Authors · 2024-08-15
>
> **Local Feature Processing** While ScaleMAE does operate both locally and globally (at the scale of an image), it embed patches by flattening their pixels and using a linear projector. We hypothesise that learning small local filters such as convolutions is a good prior (inductive bias) for the data due to its translation-equivariance. In contrast, flatten-then-project operators do not leverage this property. By “*local feature processing*”, we were referring to this prior. We will reformulate to make our reasoning clearer.
>
> **Comparison with Remote Sensing Work** As requested, we perform additional experiments related to pre-training and remote sensing work. First, we evaluate several geospatial state-of-the-art foundation models: ScaleMAE [30], DOFA [A] and USat [B]. We observe that the structural advantage of PVTv2’s architectures overtake the benefits of more relevant pre-training:
>
> | Method / Class | mIoU | Temple| Hydro| Mound| Backgnd|
> |--|--|--|--|--|--|
> | PVTv2| **52.8**| **32.3** | **39.7** | **50.7** | **89.1**  |
> | ScaleMAE|  30.4  |0.0  |16.0  | 22.8 | 82.7 |
> | DOFA| 39.6 | 13.4 | 25.9 | 33.6 | 85.5  |
> | USat | 49.6 | 28.2 | 34.0 | 48.4 | 87.7 |
>
> We also perform an additional experiment where we validate the impact of our pre-training strategy and its ability to adapt models pre-trained on RGB data to four channels (including elevation). As shown below, our approach appears necessary to meaningfully leverage the pre-training, even outperforming modern approaches such as LoRa.
>
> | Initialization | Mean | Temple | Hydro | Mound | Backgnd |
> |--|--|--|--|--|--|
> | Fully random | 46.0 | 17.9 | 32.7 | 46.1 | 87.2 |
> | Random first layer| 44.4 | 17.5 | 28.2 | 46.1 | 87.2 |
> | LoRA (rank 4)| 36.6 | 2.2 | 24.3 | 34.7 | 85.3 |
> | Proposed| **52.1** | **32.3** | **36.4** | **51.4** | **88.2** |
>
> **Influence of RGB channels** We refer the reviewer to **Table SM.1: Ablation on Channels** of the supplementary material for the requested ablation on the influence of the RGB channels.
>
> As explained on Line 129, after co-registering the ALS scans and aerial images, we concatenate the elevation and radiometric channels for each pixel. This corresponds to an early fusion scheme that jointly processes multi-source data. We acknowledge that more complex multimodal methods could also be employed, and consider this a promising avenue for future research.
>
> [A] Xiong etal. (2024). Neural plasticity-inspired foundation model for observing the earth crossing modalities.
> [B] Irvin etal (2023). USat: A unified self-supervised encoder for multi-sensor satellite imagery.

---

### Official Review · Reviewer_t7Z7 · 2024-07-24
**Significant dataset contribution for an important application with solid baseline experiments and analysis.**

**Rating:** 9
**Confidence:** 4
**Clarity:** The paper is very well-written and ea…

**Review:**

**Overall**: I highly commend the authors for their great work constructing a novel dataset for the community and running several experiments on the dataset to establish baselines. The meticulousness in both the design of the dataset and its presentation is clear.

- **Quality**:
    - Pros: The work overall is very high quality, including a clearly novel dataset for an important application that may be of high interest to the ML community, and preliminary experiments testing well-established baselines with interesting analysis.
    - Cons: There are only two major ablations. Additional ablations could help round out the experiments.
- **Clarity**:
    - Pros: The paper is very well-written and clear.
- **Originality**:
    - Pros: The work has moderate to high originality for a dataset paper, given it is the first open-access ALS archaeology dataset and the largest  among all prior datasets including closed-access ones.
    - Cons: No new algorithms were developed for the task, but this isn't necessary for a dataset paper, so not a major con.
- **Significance**:
    - Pros: The work has the potential to be a significant contribution, given the novelty of the dataset and importance of the associated task.
    - Cons: As the authors acknowledge in the limitations, the dataset only contains data for a specific civilization. Including other civilizations would increase the significance, but this is likely a substantial additional lift.

**Strengths:**

- First open access dataset for ALS archaeology, carefully designed for ML development.
- Solid set of benchmark experiments with good analysis.
- Very well-written and clear, with well-designed figures and tables.

**Additional Feedback:**

N/A

**Correctness:**

The dataset seems constructed in a sound way and the experiments designed and performed correctly.

**Documentation:**

The dataset is thoroughly documented in the Supplementary Material.

**Ethics:**

No.

**Limitations:**

The major limitations are acknowledged and discussed.

**Opportunities For Improvement:**

Candidly, it was difficult to come up with opportunities for improvement for this paper. It is very well-written and the dataset is a clear, substantial contribution to the ML and archaeology community.

1. Adding a few more experimental ablations. For example, it could be interesting to see whether pre-training improves performance (if it does, by how much). The authors comment on this briefly in the "Overall performance" section, but I think it's worth an experiment (should be straightforward to run), even if just in the Supplement.
2. It would be helpful if the authors could elaborate more in the paper on opportunities to leverage the structure of the dataset more to improve performance (and how the structure may induce novel ML approaches that have not been needed for natural images). Specifically, it seems like a larger context could be useful, but the models only input a small tile. Do the authors envision a highly-multi-scale approach could be helpful?

**Relation To Prior Work:**

The paper is well-contextualized within prior work and the proposed dataset is clearly compared to other datasets.

**Summary And Contributions:**

The paper presents a new archaeological airborne laser scanning (ALS) dataset for semantic segmentation consisting of orthophotos and LiDAR-derived normalized Digital Terrain Models annotated with 31,141 archaeological features. The authors describe the key aspects of their dataset relative to existing datasets, importantly noting that it is the first open access dataset in the field. They detail the motivation behind the dataset task, how the data is split and associated challenges they tackle, measures to prevent misuses of the dataset, and finally the dataset format and annotations. They evaluate a suite of strong semantic segmentation model baselines customized for the dataset, and discuss the quantitative results along multiple axes including the impact of the backbone architecture and the input size, as well as qualitative analysis of derived models.

---

> ### Author Rebuttal · Authors · 2024-08-15
>
> **Additional Ablations:** We thank the reviewer for the suggestion. We conducted an ablation study to evaluate the impact of our pre-training strategy and its ability to adapt models pre-trained on RGB data to four channels (including elevation). As shown below, our approach appears necessary to meaningfully leverage the pre-training, even outperforming modern approaches such as LoRa.
>
> | Initialization | Mean | Temple | Hydro | Mound | Backgnd |
> |--|--|--|--|--|--|
> | Fully random | 46.0 | 17.9 | 32.7 | 46.1 | 87.2 |
> | Random first layer| 44.4 | 17.5 | 28.2 | 46.1 | 87.2 |
> | LoRA (rank 4)| 36.6 | 2.2 | 24.3 | 34.7 | 85.3 |
> | Proposed| **52.1** | **32.3** | **36.4** | **51.4** | **88.2** |
>
> **Impact of Multi-scalarity:**  Given the scale of certain structures, such as basins, which can span several kilometers, a naive approach of simply increasing the input size and receptive fields of networks would soon encounter memory constraints. Additionally, we observed that larger context leads to severe overfitting as it significantly reduces the variety of samples. We hypothesize that a multi-scale/multi-resolution approach may overcome these limitations. We will extend the discussion in the current manuscript and explore this research direction in a future paper.
>
> >I highly commend the authors for their great work constructing a novel dataset for the community and running several experiments on the dataset to establish baselines. The meticulousness in both the design of the dataset and its presentation is clear.
>
> Thank you for your kind words; it is certainly appreciated.

---

> > ### Comment · Reviewer_t7Z7 · 2024-08-26
> >
> > Thanks for including this additional ablation and for sharing your thoughts on the potential of multi-scale/multi-resolution approaches.
> >
> > I have read through the reviewer comments and author rebuttal. A major concern raised by another reviewer was evaluation, but I agree with the authors' response to this point. Their choice of macro-average IoU is standard and appropriate, and in fact weighting by class size instead would have the opposite effect, skewing the results towards the best performing class (the majority class), as the authors state.
> >
> > I do agree with the reviewer that modifying class imbalance in the training set could be interesting to explore, although this is quite difficult to do in a segmentation setting.
> >
> > As this is a highly valuable dataset contribution with well-conducted experiments, I maintain my score.

---

### Official Review · Reviewer_ak9C · 2024-07-26
**A first publicly available dataset and benchmark for semantic segmentation of archeological sites with poor evaluation and unreliable claims.**

**Rating:** 7
**Confidence:** 4

**Review:**

This paper clearly has a lot going for it. It is the first open dataset for archeological features annotated by experts, and it evaluates state-of-the-art deep learning methods revealing that they are not well-suited for the tasks and tailormade models are required.

However, the evaluation has some severe issues which means that the claims that are made based on the results are not reliable. Also, the reproducibility of the benchmark could be improved.

Also, it is not made completely explicit what the value of the dataset is an for who especially as it is not clear how this research generalizes.

I think that this research is not yet ready for publication and hence I recommend rejecting it.

However, I believe that this research would be an important contribution to the archeological community and could spur research on segmentation tasks that require more context that state-of-the-art deep learning methods provide given that more effort is put into it.

I encourage the authors to continue this research, and I believe that the work could be influential.

**Strengths:**

Strengths:
* The first publicly available archeological dataset with expert annotations.
* The dataset is much larger than previous, closed datasets with many more annotated archeological instances.
* The figures are very nice and informative.

**Additional Feedback:**

## Generalizable:
Not clear how generalizable the dataset and benchmark are for other archeological sites because of specific historical architectures related to the Angkorian period and the terrain and geographical characteristics of Cambodia.

## Some additional comments:
* Line 44: “only open-access dataset”, please explicitly state that other, similar datasets exist in archology, but that they are closed source.
* Line 47: What is meant by misuse is not described until line 114. Please mention what it entails when it is introduced on line 47, so that the reader does not have to guess what you mean about this.
* Line 49: “ We implement rigorous safeguards …” Are they rigorous? I am not sure about the choice of words. Also, as it is not clear what the actual misuse can be (it is not yet defined), it is not clear whether the safeguards actually solve the identified potential misuses. I suggest that you clearly state the misuses that are enumerated in [10, 11] and explicitly state how your measures reduces these misuses.
* Line 53: “annotated instance”. I would like that you described which kind of archeological features have been identified and annotated. This is important information about the dataset, so please introduce the reader to these instances.
* Table 1: Sentinel 1 and 2 are satellite images while the other datasets are aerial photo imagery. Is there any significance to this? Shamsaliei et al (2024) discusses this for example.
* Line 65: Does really DeepLabv3 improve over U-net? Not according to your results. Maybe you should use another word?
* Line 102: What is a parcel?
* Line 105: How were the splits chosen to respect the global distribution of features? This should be specified in more detail.
* Line 108: Could you please provide a reference, explain what is mean with data contamination, and specify why 100 meters is enough?
* Line 110: I like the division into an out-of-distribution data set (domain shift). However, how much of a domain shift is this if the images are only spatial removed by 5km?
* Line 114: I like the measures that you have introduced to prevent misuse although I am not sure they are 100% complete or rigorous.
* Line 135: I would like to see a figure showing the class balance/imbalance.
* Line 162: It is not clear how the KALC and CALI datasets differ. I would like an analysis of this. Are the deep learning methods in any way confused by this?
* Line 195: repository => repositories.
* Line 195: “The predictions on the test set are performed along a grid …” Why? Please explain and illustrate. I am a bit confused about this.
* Line 206: Not clear to me how additional weights are initialized randomly according to a normal distribution – especially the pre-trained ones. This needs a deeper explanation and maybe an analysis of the impact.
* Line 220: “In our data, the most informative channel …” Please elaborate. I do not understand this statement.
* Line 238: “Attempts to further increase …” This is not intuitive, and it is a counterargument to the more context is better argument. Why do you think this is so? Do the images of size 512 cover the complete archeological features? If not, the context argument is false.
* Line 238: These experiments could be documented in an appendix.
* Line 274: “Despite formulating the …” Here you could provide some numbers: “Usually these models perform in the range X but here …”

**Clarity:**

Generally, the paper is well-written and structured well with nice figures. The paper is easy to comprehend. There are some smaller issues that can be improved. These are listed in the additional feedback below.

**Correctness:**

## Context:
The paper concludes that context is a significant challenge of this dataset as archeological features can span several kilometers and thus requires extensive spatial context to disambiguate (line 33). While this argument makes sense logically, the results do not necessarily convey the same message, as the top performances of the input sizes 224 and 512 are so close. The benchmark is only run once not controlling for the variation introduced by random seeds. Hence it is not clear how much results varies because of randomness, see for example Bouthillier et al (2019), Pham et al. (2020), or Zhuang et al. (2022), for discussions on variability and Gundersen et al. (2023) for a discussion on how this variability can affect conclusions.
## Reproducibility:
While both data and code are openly available, the benchmark is nor easily reproducible. The reason is that the code used for the benchmark is shared by the authors of the code, which are not the same as the authors of this paper, but the experiment code is not, nor are the weights for the models. Hence, to rerun the experiment, a third party would have to reimplement parts of the methodology (using four instead of three channels) and do hyperparameter search, which is the main bulk of the compute spent on this research. I would encourage the authors to both share the code used, the experiment setup and the wights of the models that have been used. The data including training and test set are shared to users that register on their website though.
## Class balance
The class balance is not discussed. Figure 2 indicates that the dataset has a severe class imbalance. The class balance is not investigated at all.
## Evaluation:
The choice of error metrics is not motivated. I assume that the error metrics should be weighted according to the size of the classes. The avg IoU is an average of the IoU reported in the table, which is a mean of means and thus disregards the size of the classes. This probably skews the results quite a bit toward the best performing class, which is the background class. This indicates a poor choice of metrics and hence results that cannot be trusted.


## References:

Bouthillier, X., Laurent, C., & Vincent, P. (2019, May). Unreproducible research is reproducible. In International Conference on Machine Learning (pp. 725-734). PMLR.

Gundersen, O. E., Shamsaliei, S., Kjærnli, H. S., & Langseth, H. (2023, June). On reporting robust and trustworthy conclusions from model comparison studies involving neural networks and randomness. In Proceedings of the 2023 ACM Conference on Reproducibility and Replicability (pp. 37-61).

Pham, H. V.; Qian, S.; Wang, J.; Lutellier, T.; Rosenthal, J.; Tan, L.; Yu, Y.; and Nagappan, N. 2020. Problems and opportunities in training deep learning software systems: An analysis of variance. In Proceedings of the 35th IEEE/ACM International Conference on Automated Software Engineering, 771–783

Zhuang, D., Zhang, X., Song, S., & Hooker, S. (2022). Randomness in neural network training: Characterizing the impact of tooling. Proceedings of Machine Learning and Systems, 4, 316-336.

**Documentation:**

The data is documented fairly well (in the paper) with some issues. It is a bit hard to evaluate as I do not have access to it – even though the checklists states that code and data is provided in the supplementary material.

**Ethics:**

This paper releases the first publicly available dataset of archeological findings that makes data on more than 31 000 archeological instances in Cambodia publicly available. The archeological community have been reluctant to do this previously.

The authors are aware of the issues that are associated with making such data publicly available. They have made three measures for hindering such misuse:
* Important information has been removed from the data, such as geo-referencing information.
* They have made a custom license that prohibit re-georeferencing the data and re-distributing the data.
* Access is restricted, so that those that want to work with the data must register to get access to it.

While these are good measures, I think the license also be more restrictive regarding sharing the locations of archeological sites based on this data. I think that only restriction third parties to not re-georeference the data probably is not restrictive enough.

Even though the authors have been taking measures and discussed these, which is a good thing, I suggest that the paper goes through a specialized ethics review given that I am not an expert in these issues.

**Limitations:**

The paper discusses limitations, which is nice. I have added some comments elsewhere in the review on these limitations and how I think the discussion does not cover them completely.

**Opportunities For Improvement:**

Weaknesses:
* Not clear how generalizable the findings are.
* The reproducibility could be improved.
* The claim that context is important because of how large the archeological features are not necessarily substantiated by the results.
* Evaluation: The choice of error metrics is not motivated. The error metrics should be weighted according to the size of the classes.

**Relation To Prior Work:**

The paper is missing recent and relevant literature on aerial imaging and from data-centric movement, especially regarding issues with having many experts annotating the images.

Shamsaliei et al. (2024) is another paper about semantic segmentation of aerial imaging that discusses issues such as the effect of different camera technology and the related domain shift as well as unbalanced classes and more that should be relevant for this paper.

Wirtz et al. (2024) introduces a protocol, quantitative content analysis, used in social sciences for increasing the reliability of annotations.

Aroyo and Welty (2015) discusses seven myths about human annotation. There is a growing literature related to disagreements between annotations, which probably is quite relevant for this dataset. See for example Davani, Díaz, and Prabhakaran (2022).

Shamsaliei et al. (2023) discusses how using polygons for annotating affects the performance of a semantic segmentation task of aerial images negatively. Using more detailed pixel-wise annotation methods increases performance with up to 8%.

Additional  references are given in other parts of the review.

## References:
Aroyo, L., & Welty, C. (2015). Truth is a lie: Crowd truth and the seven myths of human annotation. AI Magazine, 36(1), 15-24.

Davani, A. M., Díaz, M., & Prabhakaran, V. (2022). Dealing with disagreements: Looking beyond the majority vote in subjective annotations. Transactions of the Association for Computational Linguistics, 10, 92-110.

Shamsaliei, S., Gundersen, O. E., Alfredsen, K., & Halleraker, J. H. (2023). Towards Historical Analysis of Riverscape Development Utilizing Semantic Segmentation. Presented at the Workshop on Artificial Intelligence for Social Good, AI4SG‐23, at AAAI 2023.

Shamsaliei, S., Gundersen, O. E., Alfredsen, K. T., & Halleraker, J. H. (2024). Highlighting Challenges of State-of-the-Art Semantic Segmentation with HAIR-A Dataset of Historical Aerial Images. Journal of Data-centric Machine Learning Research.

Wirz, C. D., Sutter, C., Demuth, J. L., Mayer, K. J., Chapman, W. E., Cains, M. G., ... & Thorncroft, C. (2024). Increasing the reproducibility and replicability of supervised AI/ML in the Earth systems science by leveraging social science methods. Earth and Space Science, 11(7), e2023EA003364.

**Summary And Contributions:**

The paper presents an airborne laser scanning dataset, Archaeoscape, which spans parts of Cambodia that covers an area of 888 km2 containing 31.411 archeological instances that have been annotated by experts. The dataset is the largest of its kind and the only one that is publicly available. The paper also presents a benchmark of state-of-the-art open source deep learning semantic segmentation methods.

By making both the data and deep learning benchmark publicly available, the authors hope to encourage the archeological community to adopt open-access policies. Open-access policies have not been embraced by the community because of worries of misuse of the data for targeted looting or destruction of historical sites.

---

> ### Author Rebuttal · Authors · 2024-08-15
>
> We thank the reviewer for their very detailed review, which highlighted several areas for improvement and provided numerous highly relevant references.
>
> **Evaluation:**
> Macro-averaged IoU (unweighted class wise average) is the standard metrics used in the vast majority of semantic segmentation benchmarks in computer vision and remote sensing: see eg, S3DIS, ADE20K, CityScape, NYUDv2, PASTIS, FLAIR. Weighted IoUs favor the majority class, which is often also the easiest; in our case: background.
>
> Omitting the class frequency is indeed an oversight on our part, which we will address in the revised manuscript. Please find here the per-class frequency and weighted IoUs (wIoU) as requested.
>
> | Method / Class | Temple| Hydro| Mound| Backgnd| mIoU | wIoU |
> |--|--|--|--|--|--|--|
> | Frequency | 0.25% | 10.7% | 8.9% | 80.2% | - | - |
> | UNet | 33.3 | 32.7 | 48.6 | 87.6 | 50.5 | 78.2|
> | PVTv2| 32.3 | 39.7 | 50.7 | 89.1 | 52.8 | 80.3|
>
> **Initialization Strategy:**
> We adapted models pre-trained on RGB to 4 channels (adding the elevation) by adding new weights to the first layers and keeping all existing weights, including the ones linked to the RGB channels on the first layer. These new weights are initialized  randomly according to a normal distribution with a small variance, ensuring that the domain knowledge of the network is not destroyed by adding too much noise. We conducted below an ablation to validate this approach. Our proposed approach appear critical to leveraging the pretraining, and even beats modern approaches such as LoRa
>
> | Initialization | Mean | Temple | Hydro | Mound | Backgnd |
> |--|--|--|--|--|--|
> | Fully random | 46.0 | 17.9 | 32.7 | 46.1 | 87.2 |
> | Random first layer| 44.4 | 17.5 | 28.2 | 46.1 | 87.2 |
> | LoRa| 36.6 | 2.2 | 24.3 | 34.7 | 85.3 |
> | Proposed| **52.1** | **32.3** | **36.4** | **51.4** | **88.2** |
>
>
> **Generalizability:**
> The models trained on Archaeoscape are not expected to generalize to the broader task of universal archaeological segmentation. As mentioned in our limitations section, our dataset is specific to the Khmer culture, and the model is not expected to perform well in detecting, for instance, Viking burial mounds or Mayan Aguadas.
>
> It is, however, a valid question whether our conclusions regarding deep learning architectures could apply to other regions and cultures, particularly those lacking monumental features. Unfortunately, since none of the existing ALS archaeological datasets are open-source, we are unable to verify this experimentally. We hope that our work will lead the way towards more openness regarding archaeological data.
>
> **Reproducibility: Experiment Setup:**
> The code, scripts, and weights necessary to reproduce all experiments will be made available through an open-access repository. Potential users are not required to conduct their own training and hyperparameter tuning unless they choose to, as the checkpoints of all trained models will be provided.
> Regarding access to the dataset, reviewers can use the [url](https://archaeoscape.ai/reviewer_access) and the password : **psychic_hexagon_eternal_sloped_acclimate_precise_pony_accumulator_clip** provided in the **SM.6-Reviewer Access** section of the supplementary material. If any issues arise with this method, please let us know, and we will arrange an alternative way to share the dataset.
>
> **Reproducibility: Variance Estimation:**
> We have re-trained our best model (PVTv2) 5 times with different seeds and observed an empirical variance slightly below 0.5%. We will add this analysis in the manuscript, as well as the highly relevant references provided by the reviewer.
>
> **Importance of Large Context:**
> Images of size 256 pixels (128m) or 512 pixels (256m) are still much smaller than the typical size of hydrology basins which can reach several kilometers, as shown in Fig4 (a), bottom row. This is why the performance of these networks, while higher for hydrology, remains relatively low. We have tried larger contexts and rapidly run into issues of overfitting, as the diversity of samples decreases. We will add a more in depth discussion on the subject.
>
> **Annotation Quality:**
> We thank the reviewer for the insightful references on handling noisy annotation quality. We highlight that archaeological annotations differ from typical computer vision annotations in the high expertise required, and the numerous field operations (over 800 for Archaeoscape over 10 years) necessary to confirm the presence of vestiges.
>
> **Split Stratification:**
> The splits were chosen such that they all contain representative features (large-scale hydraulic engineering sites, monumental temples, subtle features) and terrain types (dense and scarce occupation, hills, floodplains). More detail will be added in the supplementary. Here we provide below the class distribution per split.
>
> | Split | Temple| Hydro| Mound| Backgnd|
> |--|--|--|--|--|
> | Overall| 0.25% | 10.7% | 8.9% | 80.2% |
> | Train| 0.23% | 9.7% | 8.4% | 81.7% |
> | Val| 0.25% | 20.6% | 10.8% | 68.4% |
> | Test| 0.34% | 8.9% | 9.5% | 81.2% |
>
> **Privacy:**
> The locations and raw data of Archaeoscape tiles are never distributed: all annotations, terrain models, and images are centered on (0,0,0). We refer the reviewer to our response to ethics reviewers for more details.

---

> > ### Author Rebuttal · Authors · 2024-08-15
> >
> > **Additional comments:**
> > We thank the reviewer for their meticulous reading of our manuscript. We have incorporated all their suggestions in the revised manuscript. We provide here the information requested.
> > - S1/S2 images have higher spectral resolution than our aerial orthophotography (12 bands vs 4) but lower spatial resolution (10m vs 50cm). The spectral resolution is typically useful for classifying vegetation, and is thus not highly relevant for detecting structures that are often under the canopy. As shown in our experiment in **Table SM.1** of the supplementary materials, the RGB channels have a limited impact on performance.
> > - Parcels are specific areas with clear boundaries, that we divide the dataset into as explained line 102. We will add a mention to this word in the caption of Fig2 for clarity.
> > - Data contamination refers to the spatial self-correlation of the data, which could lead to models implicitly using training data for classifying test data. For the test set, we use two parcels that are in different areas to eliminate all contamination. We also use two parcels within the same areas but with a 100m buffer, thus limiting contamination. A larger buffer may work even better, but would remove a lot of annotations from the dataset.
> > - Each site exhibits unique idiosyncrasies in its structure and hydrology. Within our area of coverage, we have settlements dating from the 8th to the 13th centuries, which introduces a significant domain shift between different sites. However, all the data pertain to the same Khmer culture.
> > - Inference on the validation and test set is performed by taking crops along a regular grid with overlap, a standard practice for segmenting large geospatial data [A,B]
> > - Elevation is the most informative channel, as it alone can lead to good results. See **Table SM.1: Ablation on Channels** of the supplementary for an ablation.
> > - The [KALC](https://www.pnas.org/doi/full/10.1073/pnas.1306539110) and [CALI](https://cordis.europa.eu/project/id/639828/results) projects performed multiple acquisition campaigns across 10 years. The hardware and actors involved are largely similar, and there is no distinctive difference between their acquisitions. The main domain shifts between parcels are determined by the sites they have been extracted from, and not from the origin of their funding.
> >
> > [A] Li etal (2022) Full convolution neural network combined with contextual feature representation for cropland extraction from high-resolution remote sensing images. Remote Sensing.
> > [B] An etal (2020). Overlap training to mitigate inconsistencies caused by image tiling in CNNs. ICITAAI.

---

> > > ### Comment · Reviewer_ak9C · 2024-08-27
> > >
> > > Thank you for your detailed feedback and revising the manuscript according to the comments. I have adjusted my score reflecting this.

---

### Official Review · Reviewer_puKz · 2024-08-03

**Rating:** 9
**Confidence:** 3
**Correctness:** Yes
**Clarity:** Yes

**Review:**

This paper is clearly written, and the contribution is significant to the community. It improves on the next largest dataset in the field by a factor of 4 with 6 times of labels - and is the only proposed dataset with an open access license. An effort has been invested into ensuring that the data cannot be misused, such as a custom license, and the removal of georeferencing elements. One main concern that is also highlighted by the authors is the cultural specificity (focusing on Khmer) however, I believe that the size and quality of the dataset and labels is likely helpful enough to advance other ALS tasks given the lack of currently existing open-access data. Overall this was a very enjoyable paper to read.

**Strengths:**

- First open-access and high-res dataset in ALS archeology
- 4x larger than the next dataset with quality-ensured labelling from expert archeologists (6x more labels than the previous largest dataset)
- An effort was invested into preventing misuse of the data
- A benchmark is provided demonstrating that opportunities for the ML community to get involved into ALS archeology
- Providing a dataloader and tooling

**Additional Feedback:**

N/A

**Documentation:**

Yes

**Ethics:**

Yes

**Limitations:**

- Privacy risk and data misuse is still possible, despite efforts undertaken such as removal of georeferencing element and pseudo naming
- Focused solely on Khmer archeology and a unique geographic region

**Opportunities For Improvement:**

- Provide a documentation/section in the paper on the dataloader
- Currently the dataset is in raw geospatial format (raster and .gpkg). One could think of providing a larger version of the dataset that comes already preprocessed ML-ready format (e.g. COCO, image folders)

**Relation To Prior Work:**

Yes

**Summary And Contributions:**

This paper introduces Archaeoscape, a large-scale archaeological airborne laser scanning (ALS) dataset spanning 888 km2 in Cambodia with 31,141 annotated archaeological features from the Angkorian period. The dataset is the first ALS archeology dataset that is open-access and high-res (0.5m/px). The authors established a semantic segmentation benchmark on the novel dataset using CNN, ViT and HViT models and concluded that these models struggle to achieve high scores, attributing this poor performance to the unique challenges of ALS archaeology.

---

> ### Author Rebuttal · Authors · 2024-08-15
>
> **ML-Ready Version:** Due to the relatively small size of the dataset, splitting the tiles into images directly would result in a relatively small number of images (888 km² / (128 m * 128 m) ~ 50k images), which could lead to overfitting. Our data loader dynamically takes random crops with random location, orientation, and small scale changes, resulting in a much richer dataset than a regular tiling would. We have ensured that this processing is transparent, and the data loader loads a batch of images in a deep learning-ready format, unaffected by the complexity of the sampling.
>
> However, we can easily comply with this  request. If the reviewers think it would be useful for computer vision researchers, we can create and share a bank of random crops (e.g., 250k images) to be used with a classic data loader. We await your response to this rebuttal to proceed.
>
> **Data Loader Documentation:** The GitHub repository documentation https://github.com/archaeoscape-ai/archaeoscape will explain in detail how the data loader works. However, we agree that including key elements of its mechanism in the supplementary material is a good idea. We will add this in the revised manuscript.
>
> > Overall this was a very enjoyable paper to read.
>
> Thank you! This is the best compliment we could hope for.

---

### Author Rebuttal · Authors · 2024-08-15

## Ethics Reviews ##

We thank the ethics reviewers for their valuable feedback and recommendations to make the Archaeoscape data as safe and respectful as possible. Below, we outline our ongoing efforts and future plans to promote the ethical and responsible use of this dataset.

**Ethical Compliance**

Archaeoscape data acquisition was supported by two ERC Grants: [CALI, #639828](https://cordis.europa.eu/project/id/639828) and [archaeoscape.ai, #866454](https://cordis.europa.eu/project/id/866454). Comprehensive ethical reviews have been conducted to ensure our work aligns fully with EU standards for ethics and cultural sensitivity.

**License Agreement Updates**

Beyond our existing precautions, which prohibit georeferencing and commercial use, we will be updating the license agreement with an additional clause. This clause must be agreed upon by all users before access to the data is granted.

> By agreeing to this form, you commit to the following:
> - **Usage Restrictions:** Archaeoscape must not be used for excavation, digs, removal of cultural artifacts, or any activities that could directly impact the environment.
> - **Respect for Cultural Heritage:** The Archaeoscape dataset must be used in a manner that respects the cultural heritage of the Khmer civilization and associated cultures. Any activities that could harm or exploit these cultural sites or the environment are strictly prohibited.

**Local Collaboration**

Our focus on archaeological sites related to the Khmer civilization necessitates careful consideration of cultural implications. Several authors of the paper are members of the EFEO (The French School of Asian Studies), leading experts in archaeology and its ethical implications (see eg: [Ethics in Archaeological Lidar](https://journal.caa-international.org/articles/10.5334/jcaa.48)).

The EFEO operates a unique network of 18 field research centers across 12 countries in Asia, working closely with local authorities and stakeholders under officially signed Memorandums of Understanding (MoUs). Our efforts are conducted in collaboration with relevant local partners, with a strong emphasis on preventing any form of cultural appropriation or exploitation, with both LiDAR acquisition campaigns  approved and guided by His Excellency Dr Sok An, Deputy Prime Minister and Chairman of the Council of Ministers, and Chairman of the APSARA National Authority. Our on-the-ground presence enables us to continuously monitor for potential misuse and, if necessary, update the terms of the license or the modalities of data sharing.

**Public Outreach**

We are deeply committed to outreach activities, including organizing seminars with professionals from the APSARA National Authority and the Ministry of Culture and Fine Arts of Cambodia, as well as public lectures and workshops for other involved local partners. A full copy of the LiDAR data acquired has been provided to local authorities, and we have organized a series of training workshops and courses. These efforts are aimed at building a community of proficient users within Cambodian government agencies responsible for overseeing national cultural heritage, ensuring that LiDAR products are seamlessly integrated into their planning and decision-making processes.

These initiatives have fostered a collaborative environment, and we believe that the creation of the Archaeoscape dataset plays a crucial role in further empowering local and national authorities to manage cultural properties more effectively, mitigate environmental impacts, and implement best practices in sustainable archaeology.

---

### Decision · Program_Chairs · 2024-09-26

**Decision:**

Accept (Spotlight)

**Comment:**

The article introduces Archaeoscape, a large-scale open-source dataset designed to leverage deep learning techniques for analyzing airborne laser scanning (ALS) data and addressing challenges in traditional archaeological methods. While the Archaeoscape dataset holds significant potential for advancing archaeology, the article's benchmarking and experimental evaluations are insufficient and need more thorough testing. All four reviewers recognize the novelty and recommend acceptance.